# The *Arabidopsis* MYB96 Transcription Factor Mediates ABA-Dependent Triacylglycerol Accumulation in Vegetative Tissues under Drought Stress Conditions

**DOI:** 10.3390/plants8090296

**Published:** 2019-08-22

**Authors:** Hong Gil Lee, Mid-Eum Park, Bo Yeon Park, Hyun Uk Kim, Pil Joon Seo

**Affiliations:** 1Department of Chemistry, Seoul National University, Seoul 08826, Korea; 2Department of Bioindustry and Bioresource Engineering, Plant Engineering Research Institute, Sejong University, Seoul 05006, Korea; 3Department of Technology Dissemination, Agricultural Technology Center, Gwangyang 57737, Korea; 4Plant Genomics and Breeding Institute, Seoul National University, Seoul 08826, Korea

**Keywords:** abscisic acid, *Arabidopsis*, MYB96, triacylglycerol, drought tolerance

## Abstract

Triacylglycerols (TAGs), a major lipid form of energy storage, are involved in a variety of plant developmental processes. While carbon reserves mainly accumulate in seeds, significant amounts of TAG have also been observed in vegetative tissues. Notably, the accumulation of leaf TAGs is influenced by environmental stresses such as drought stress, although underlying molecular networks remain to be fully elucidated. In this study, we demonstrate that the R2R3-type MYB96 transcription factor promotes TAG biosynthesis in *Arabidopsis thaliana* seedlings. Core TAG biosynthetic genes were up-regulated in *myb96-ox* seedlings, but down-regulated in *myb96-*deficient seedlings. In particular, ABA stimulates TAG accumulation in the vegetative tissues, and MYB96 plays a fundamental role in this process. Considering that TAG accumulation contributes to plant tolerance to drought stress, MYB96-dependent TAG biosynthesis not only triggers plant adaptive responses but also optimizes energy metabolism to ensure plant fitness under unfavorable environmental conditions.

## 1. Introduction

Plants synthesize TAGs for carbon and energy storage [1]. Biosynthesis of TAG occurs in endoplasmic reticulum (ER) by a set of membrane-associated enzymes [2]. Fatty acid chains are transferred from acyl-CoA to the glycerol-3-phosphate backbone at *sn*-1 and *sn*-2 positions through the acyltransferase reactions of glycerol-3-phosphate acyltransferase (GPAT) and lysophosphatidic acid acyltransferase (LPAT), respectively [3,4]. Lysophosphatidic acid at the *sn*-3 position is dephosphorylated by phosphatidate phosphatase, forming diacylglycerol (DAG) [5]. A third fatty acid is then transferred to the *sn*-3 position of DAG by diacylglycerol acyltransferase (DGAT) [6].

Two major DGAT families that have no homology to one another, DGAT1 and DGAT2, have been identified in higher eukaryotes [7,8]. The *Arabidopsis thaliana* genome has one *DGAT1* (At2g19450) and one *DGAT2* (At3g51520). The DGAT1 protein plays a major role in seed oil accumulation, whereas the DGAT2 enzyme is important for unusual fatty acid accumulation [8,9]. Particular emphasis has been placed on the DGAT1 protein, because it is considered the rate-limiting enzyme in *Arabidopsis* TAG biosynthesis [6,10,11].

Although DGAT1 is an important enzyme for TAG biosynthesis, other enzymes also act in parallel for TAG biosynthesis. The *sn*-2 acyl group of phospholipids, such as phosphatidylcholine (PC) and phosphatidylethanolamine (PE), can be transferred to the *sn*-3 position of DAG [12,13]. This reaction is catalyzed by phospholipid:diacylglycerol acyltransferase (PDAT) [13,14]. Two putative *PDAT* genes have been identified in *Arabidopsis* [13], but only *PDAT1* (At5g13640) has been intensively investigated [14]. While *PDAT1*-deficient mutants exhibit no substantial alterations in lipid and fatty acid contents [14,15], disruption of both *DGAT1* and *PDAT1* genes leads to a 70–80% decrease in seed TAG contents, indicating that DGAT1 and PDAT1 have complementary functions in *Arabidopsis* TAG biosynthesis [6].

TAG is primarily synthesized in seeds, but significant amounts of TAG also accumulate in vegetative tissues under environmentally unfavorable conditions. For instance, galactolipids and phospholipids are substantially converted into TAG in leaves during senescence and under stress conditions [16,17,18,19,20]. In addition, *DGAT1* expression is induced in vegetative tissues in response to osmotic and low-nitrogen stresses and abscisic acid (ABA) treatment [21].

Transcriptional regulation is a crucial molecular scheme in TAG biosynthesis in *Arabidopsis*. WRINKLED1 (WRI1), an APETALA2 (AP2)/ethylene-responsive element-binding protein (EREBP) transcription factor, is a representative regulator of de novo fatty acid biosynthesis and contributes to TAG accumulation by directly binding to the promoters of glycolytic and fatty acid biosynthetic genes, such as *PKp-β1* and *BCCP2* [22,23,24,25]. Accordingly, *wri1* mutations result in an approximately 80% decrease in seed TAG levels [22,26]. The AP2/EREBP-type ABA-INSENSITIVE 4 (ABI4) transcription factor is also important for TAG biosynthesis particularly under stress conditions, possibly by directly activating the *DGAT1* gene [21,27]. In addition, MYB96 has been characterized as a mater regulator that facilitates transcriptional control of *DGAT1* and *PDAT1* to ensure TAG accumulation in seeds [28].

The MYB96 transcription factor is a key regulator of ABA signaling and mediates a variety of plant responses to ABA, such as drought tolerance, lateral root development, and cuticular wax biosynthesis [29,30,31]. Here, we report that MYB96 is also involved in TAG biosynthesis in vegetative tissues through the transcriptional regulation of *DGAT1* and *PDAT1*. The *MYB96*-overexpressing activation-tagging line (*myb96-ox*) showed a substantial increase of TAG accumulation, whereas ABA-induced TAG accumulation was reduced in the *MYB96*-deficient mutant seedlings. TAG levels are unequivocally associated with plant tolerance to environmental stresses, as exemplified by reduced tolerance of TAG-deficient mutants to drought stress. Together, our findings indicate the biological relevance of TAG accumulation in stress adaptation and provide an insight into how TAG biosynthesis is comprehensively regulated under adverse environmental conditions.

## 2. Results

### 2.1. TAG Accumulation is Increased in myb96-ox Seedlings

The ABA-inducible MYB96 transcription factor regulates diverse aspects of plant developmental and metabolic processes to enhance plant fitness and adaptation under environmental stress conditions, including lateral root development, stomatal opening, anthocyanin accumulation, and cuticular wax biosynthesis [29,30,31]. Notably, ABA stimulates TAG biosynthesis in leaves [21]. Given that MYB96 promotes expression of *DGAT1* and *PDAT1* in seeds [28], we supposed that MYB96 may also mediate ABA-inducible TAG biosynthesis in vegetative tissues.

DGAT1 and PDAT1 are key rate-limiting enzymes in TAG biosynthesis [6,10,11]. To assess the connection between MYB96 and TAG biosynthesis in vegetative tissues, total TAG content in seedlings was measured. A substantial increase in TAG accumulation was observed in the *myb96-ox* seedlings compared to wild-type seedlings, while a reduction of TAG levels in the *myb96-1* mutant were not obvious (Figure 1 and Appendix A). Levels of TAG in *myb96-ox* seedlings were comparable to TAG levels in wild-type seeds (Figure 1 and Appendix A). These results suggest that MYB96 positively regulates TAG biosynthesis in vegetative tissues, likely through promoting expression of *DGAT1* and *PDAT1*.

### 2.2. ABA- and Stress-Induced Expression of TAG Biosynthesis Genes Requires MYB96

MYB96 plays an essential role in mediating ABA signaling. Given that ABA stimulates TAG accumulation in *Arabidopsis* leaves [21], it is plausible that the MYB96 transcription factor is involved in this process. To test this possibility, we analyzed effects of ABA on transcript accumulation of TAG metabolic genes. Among the genes examined, *DGAT1*, *DGAT2*, *DGAT3*, *PDAT1*, *FAE1*, *FAD2*, *FAD3*, and *LPCAT1* were induced by ABA in seedlings (Appendix A). Notably, the expression of rate-limiting TAG biosynthetic genes, *DGAT1* and *PDAT1*, was specifically dependent on MYB96 (Figure 2A and Appendix A). The two genes were regulated by ABA with similar induction kinetics in wild-type seedlings. Transcript accumulation of them increased continuously with time in response to exogenous ABA treatment (Figure 2A). However, induction of their expression by ABA was diminished in the *myb96-1* mutant (Figure 2A). Similarly, *DGAT1* and *PDAT1* were also induced upon exposure to osmotic stress and dehydration in a MYB96-dependent manner (Figure 2B–D). In accordance with this, osmotic stress induction of *DGAT1* and *PDAT1* was also impaired in ABA-deficient *aba3-1* mutant (Appendix A), supporting the intimate role of MYB96 in transcriptional activation of *DGAT1* and *PDAT1* upon the ABA accumulation.

To further support MYB96 regulation of TAG biosynthesis in the presence of ABA, TAG contents were measured in wild-type and *myb96-1* seedlings that were treated with 10 μM ABA. In the presence of ABA, TAG levels were significantly elevated in wild-type plants, while the TAG accumulation was impaired in *myb96-1* (Figure 3A,B). In contrast, *myb96-ox* further increased TAG levels (Figure 3A,B). Though *myb96-ox* seedlings exhibited stunted growth and dwarfism, which might influence TAG levels, these results indicate that MYB96 is an unequivocal positive regulator of TAG biosynthesis under environmental stress conditions.

### 2.3. TAG-Deficient Mutant is Sensitive to Drought Stress

The ABA-signaling mediator MYB96 is known to confer drought stress tolerance [29]. Since MYB96 stimulates TAG biosynthesis, particularly in the presence of ABA, TAG accumulation might be associated with plant adaptation to environmental stress [21,32]. To investigate this hypothesis, we employed the *dgat1-1* (147 bp insertion at the central region of intron 2) and another *dgat1* mutant allele *dgat1-2* (T-DNA insertion in the last exon) with reduced TAG contents [10,11,33] and examined their adaptation capability to drought stress. Notably, the *dgat1*-deficient mutants were more susceptible to drought stress than wild-type plants (Figure 4A). Survival rate analysis showed that approximately 60% of wild-type plants survived, whereas only 20% of *dgat1* mutant plants were tolerant to drought stress after two weeks of water deficit (Figure 4B). As a comparison, the *wri1-3* mutant plants, which show lipid metabolic defects specifically in seeds [24,25,34], did not exhibit any distinguishable phenotypes compared with wild-type plants under water-deficit conditions (Appendix A), supporting the finding that TAG accumulation in vegetative tissues is required for drought tolerance.

To confirm that MYB96-mediated TAG accumulation is relevant in drought tolerance, we genetically crossed a 35S:*MYB96* transgenic plant with *dgat1* mutants and measured its survival rate upon exposure to drought stress. The higher drought tolerance of 35S:*MYB96* transgenic plants was significantly compromised, but not completely, by introduction of the *dgat1* mutation (Figure 4C,D). The partial suppression of drought tolerance in 35S:*MYB96*/*dgat1* might be due to either multiple target traits regulated by MYB96 [29,30,31], or enhanced compensation of reduced DGAT1 activity in 35S:*MYB96-MYC* transgenic lines by PDAT1, as shown in elevated *PDAT1* expression in *dgat1* mutants (Appendix A). Taken together, these results indicate that MYB96-dependent activation of TAG biosynthesis leads to plant fitness under drought conditions (Figure 5).

## 3. Discussion

It has been demonstrated that TAG biosynthesis is closely associated with ABA signaling. ABA plays a fundamental role in TAG accumulation during seed development and maturation, which is intimately associated with seed dormancy and germination [35,36]. In addition, TAG accumulation is also observed in vegetative tissues, albeit with its low levels relative to seeds. While TAG levels are marginally changed by developmental stages and sugar applications in leaves [37,38,39], TAG accumulation is moderately increased in response to ABA, nitrogen deficiency, and osmotic and oxidative stresses in vegetative tissues [21,27]. In *Arabidopsis* seedlings, reduced TAG biosynthesis leads to hypersensitivity to various abiotic stresses, whereas increasing lipid droplets alleviate the damages caused by environmental stresses [40,41]. Furthermore, TAG-derived fatty acids are also involved in guard cell movement [42,43].

ABA-inducible TAG accumulation in vegetative tissues is mainly mediated by the trio of MYB96, DGAT1, and PDAT1. DGAT1 is most likely a central component responsible for ABA-induced TAG accumulation in leaves, whereas PDAT1 might play a supplemental role in this process [44]. MYB96 coordinates expression of two core TAG biosynthetic genes and ensures proper levels of TAG biosynthesis in any given condition. Under drought conditions, MYB96 would primarily depend on *DGAT1* for TAG biosynthesis, and also regulate *PDAT1* to properly supplement DGAT1 activity. Accordingly, *dgat1* mutants are susceptible to water deficit, and introduction of *dgat1* mutations into 35S:*MYB96-MYC* results in partial reduction of drought tolerance of 35S:*MYB96-MYC*.

TAG accumulation in vegetative tissues is particularly important because it enhances plant fitness under environmental stress conditions. Although it is currently unclear how TAG regulates plant adaptability, several possibilities are suggested. The chemical composition of cellular membranes is an important factor for eliciting plant responses to abiotic stresses [45]. For instance, cold stress leads to reduced membrane fluidity, and the remodeling of physico-chemical membrane properties triggers rapid responses to temperature changes [46]. Recycling of TAGs to produce fatty acids may influence the homeostasis of fatty acid levels in cellular membranes to elicit adaptive responses. Alternatively, TAG-induced changes in membrane properties may contribute to maintaining cellular membrane integrity under adverse stress conditions.

TAG is a major form of carbon storage. Under stress conditions, energy reserves ensure plant growth and development with limited photosynthetic activity. In this regard, ABA-inducible TAG accumulation is important for long-term stress acclimation in plants. Indeed, sugar content and hexokinase activity are significantly altered in *dgat1* mutant seedlings, and *dgat1* mutants are accordingly sensitive to ABA and osmotic stress during germination and post-germinative seedling growth [47].

Altogether, MYB96 regulates TAG biosynthesis to enhance plant adaptive capability to environmental stress. This transcription factor not only triggers drought-related traits, such as stomatal closure, lateral root inhibition, and cuticular wax accumulation [29,30,31], but also regulates carbon and energy storage to further ensure plant growth and development under long-term stress conditions.

## 4. Materials and Methods

### 4.1. Plant Materials and Growth Conditions

All experiments were performed using *Arabidopsis thaliana* (Columbia-0 ecotype), unless specified otherwise. Seeds were stratified at 4 °C for 2 days and subsequently germinated and grown under long-day (LD) conditions (16-h light/8-h dark cycles) at 22–23 °C. The *myb96-1* (GABI 120B05) T-DNA insertional knock-out mutant and *myb96-ox* activation-tagging overexpressing mutant have previously been reported [29]. To generate 35S:*MYB96* transgenic plants, the full-length *MYB96* cDNA was cloned into MYC-pBA002 binary vector. The *dgat1-1* (AS11, CS3861), *dgat1-2* (A7, SALK 039456), and *wri1-3* (SALK 085693) mutants have also previously been described [11,24,33,48]. Gene expression in mutant and transgenic plants was verified by reverse transcription (RT)-PCR before use.

### 4.2. Quantitative Real-Time RT-PCR Analysis

Plant tissue was homogenized in liquid nitrogen. The homogenized samples were mixed with equal volume of TRI agent (TAKARA Bio, Singa, Japan), and total RNA was extracted. Extracted RNAs were treated with DNase I at 37 °C, and first-strand cDNA was synthesized from 2 μg of RNA using the Moloney Murine Leukemia Virus (M-MLV) reverse transcriptase (Dr. Protein, Seoul, Republic of Korea) and oligo (dT18).

For qPCR, cDNAs were prepared with TOPreal qPCR 2X PreMIX (SYBR Green with low ROX) (Enzynomis, Seoul, Republic of Korea). Real-time (RT)-PCR was performed on the Step-One Plus Real-Time PCR System (Applied Biosystems, Foster City, CA, USA). The PCR primers used are listed in Appendix A. The values for each set of primers were normalized relative to the *EUKARYOTIC TRANSLATION INITIATION FACTOR 4A1* (*eIF4A*) gene (At3g13920). To compare the transcript levels between different samples, the 2^−∆∆CT^ method was used. The threshold cycle (C_T_) was automatically determined for each reaction by the system set with default parameters. Individual experiments were repeated twice with three expression quantifications being performed for each sample, and standard deviation was evaluated for each time-point analyzed for gene expression. The melt curve analysis for the products amplified by the PCR reaction was performed.

### 4.3. TAG Determinations

For TAG measurement, two-week-old seedlings grown on half-strength Murashige and Skoog (MS) medium (without sucrose, pH 5.7) were used. TAG was detected with thin layer chromatography (TLC) analysis. Deep-frozen seedlings in liquid N_2_ were grinded by tissuelyser II (Qiagen, Qiagen, Hilden, Germany) and treated with a 1-mL cold solution mixture of 10:10:1 (*v/v*) chloroform–methanol–formic acid for 12 h at −20 °C. The samples were separated by centrifugation at 20,000× *g* for 10 min. Supernatants were transferred to a new tube, and the remaining pellets were re-extracted in 0.5 mL of 5:5:1 (*v/v*) chloroform–methanol–water by centrifugation at 20,000× *g* for 10 min. The second supernatants were combined with stored first supernatants, mixed with 0.41 mL of solution (1 M KCl, 0.2 M H_3_PO_4_), and centrifuged at 20,000× *g* for 10 min. The separated bottom lipid layer was transferred to a new tube and lyophilized with N_2_ gas. Lipids dissolved in 20 μL of chloroform were spotted on TLC plates (silica gel G60 20 × 20 cm plates; EM Separations Technology), and the spots were developed with hexane–diethylether–acetic acid (140:60:2 by vol.). Lipids on TLC were visualized by staining with iodine. TAG amounts were measured using the sum of fatty-acid methyl esters derived from TAGs with glyceryl triheptadecanonic acid (17:0–TAG, Sigma, St. Louis, MO, USA) as an internal standard, using gas chromatography (GC) analysis as described [37].

### 4.4. Treatments with ABA and Drought Stress

Ten-day-old seedlings grown under LD conditions at 22–23 °C were used for treatment with exogenous ABA and osmotic stress. The seedlings were transferred to MS–liquid medium supplemented with 20 μM (+)-*cis,trans*-ABA (L06278) (Alfa Aesar, Ward Hill, MA, USA), 150 mM NaCl, or 150 mM Mannitol. For drought treatment, two-week-old plants were dried by halting watering for two weeks. We measured survival rate after rehydration. Three independent biological measurements with at least 30 plants in each set were averaged.

### 4.5. Accession Numbers

Sequence data from this article can be found from the *Arabidopsis* Genome Initiative or GenBank/EMBL databases under the following accession numbers: *DGAT1* (At2g19450), *DGAT2* (At3g51520), *DGAT3* (At1g48300), *FAD2* (At3g12120), *FAD3* (At2g29980), *FAE1* (At4g34520), *GPAT9* (At5g60620), *GPDHc1* (At2g41540), *LPAT2* (At3g57650), *LPCAT1* (At1g12640), *LPCAT2* (At1g63050), *PDAT1* (At5g13640), *PDAT2* (At3g44830), *PDCT* (At3g15820), *WRI1* (At3g54320), and *MYB96* (At5g62470).

## Figures and Tables

**Figure 1 plants-08-00296-f001:**
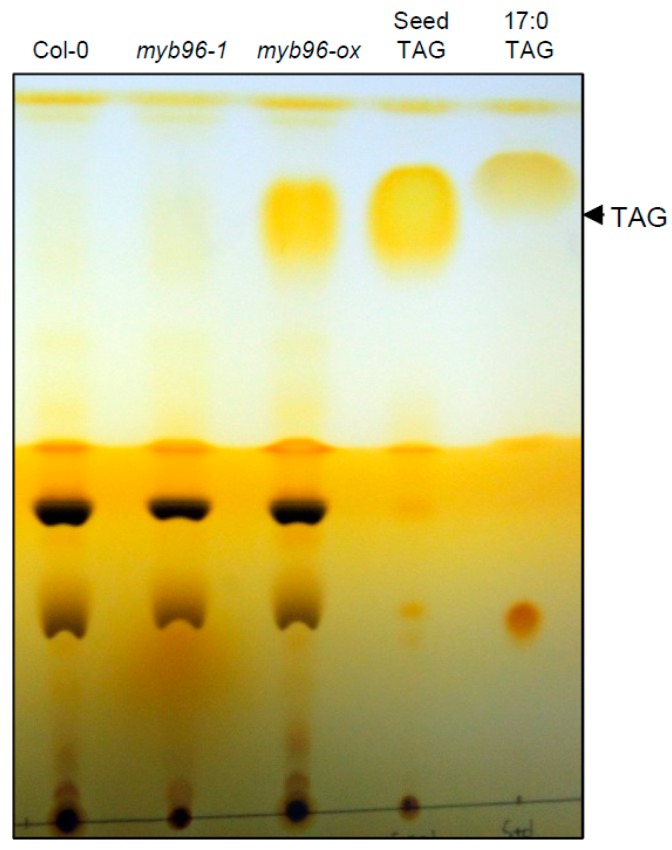
TAG accumulation in *myb96-ox* seedling. Ten-day-old seedlings grown under long-day (LD) conditions were used to extract total lipids. Extracted lipids were separated in thin layer chromatography (TLC) plates. Three independent biological replicates were analyzed, and the representative image is shown. TAG from wild-type seeds and 17:0 TAG standard were loaded on the right of the plate to indicate positions of the lipids.

**Figure 2 plants-08-00296-f002:**
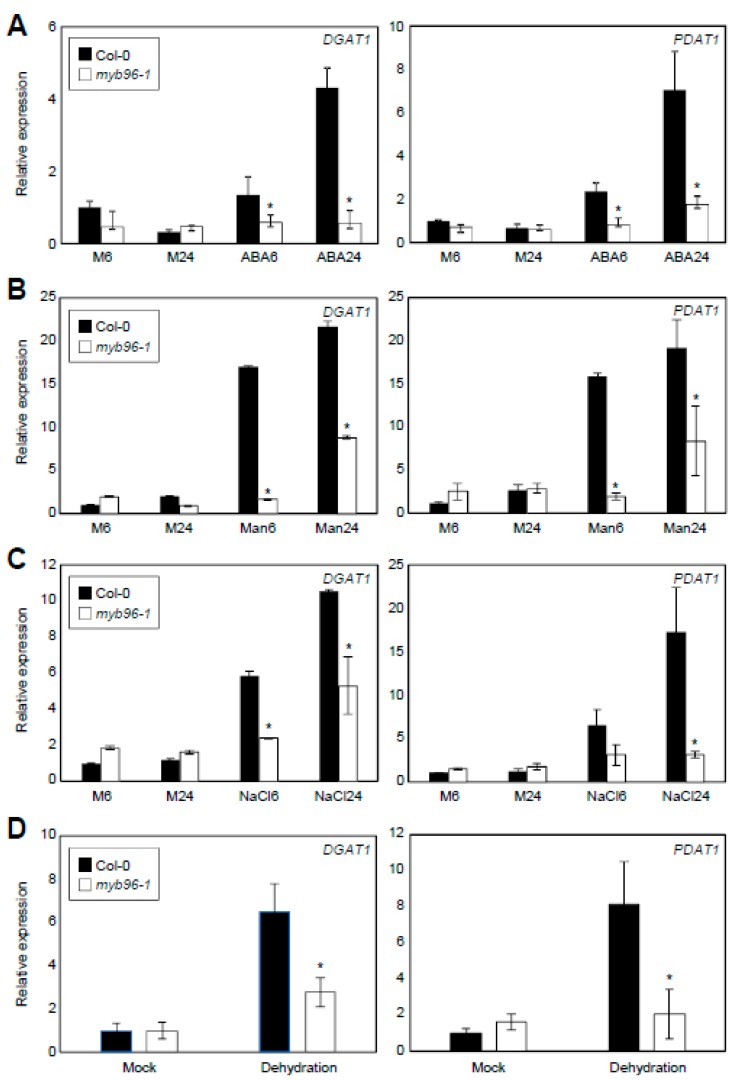
Compromised expression of *DGAT1* and *PDAT1* in *myb96-1* upon exogenous ABA treatment. Ten-day-old seedlings grown under LD conditions were transferred to Murashige and Skoog (MS)–liquid medium supplemented with or without 20 μM ABA (**A**), 150 mM mannitol (Man) (**B**), or 150 mM NaCl (**C**) and incubated for the indicated time period (h). For dehydration treatment, 14-day-old seedlings were air-dried for 2 h (**D**). Transcript accumulation of *DGAT1* and *PDAT1* was analyzed by real-time quantitative PCR (RT-qPCR). The *Y*-axis represents relative expression levels normalized by the *EUKARYOTIC TRANSLATION INITIATION FACTOR 4A1* (*eIF4a*) gene (At3g13920). Three independent experiments with three biological replicates were averaged. Statistically significant differences between wild-type and mutants at corresponding time points are indicated by asterisks (Student’s *t*-test, * *P* < 0.05). Bars indicate the standard error of the mean. The bars labeled Mock represent the background measured in untreated plant samples. M, mock.

**Figure 3 plants-08-00296-f003:**
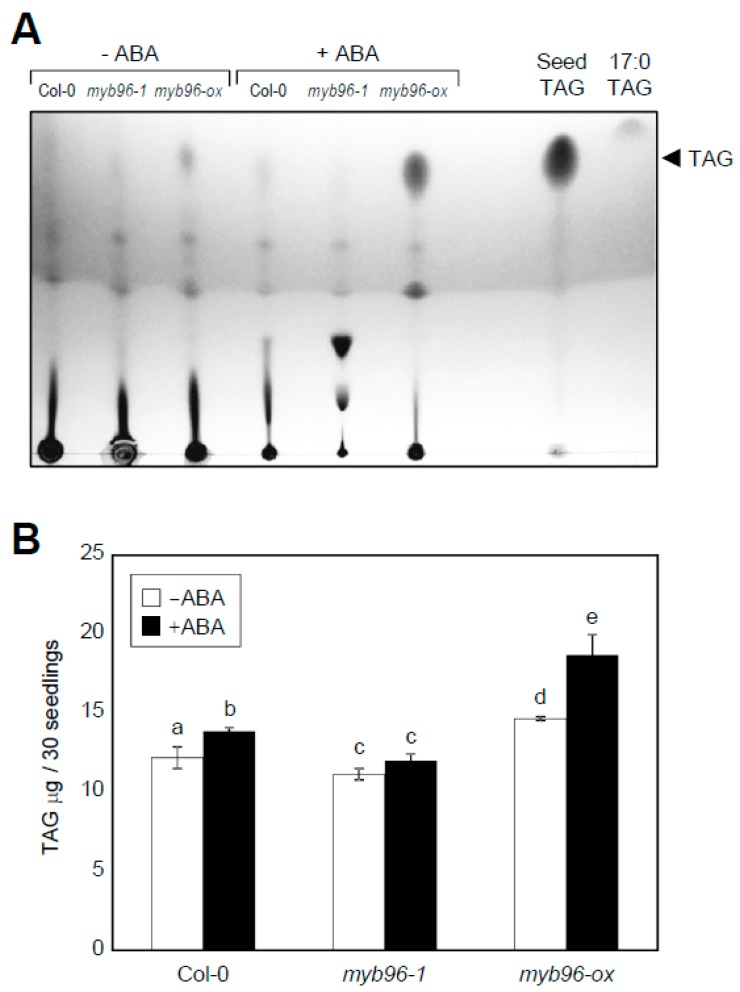
Effects of ABA on TAG accumulation in *myb96-ox* and *myb96-1*. Five-day-old seedlings grown under LD conditions were transferred to MS–medium supplemented with 10 µM ABA and incubated for additional 2 days. Total lipids were extracted and separated in TLC plates. (**A**) Separation of total lipids by TLC. Three independent experiments with three biological replicates were analyzed, and representative images are shown. TAG from wild-type seeds and 17:0 TAG standard were loaded on the right of the plate to indicate positions of the lipids. (**B**) Quantification of TAG by gas chromatography (GC) analysis. Three independent experiments with three biological replicates were averaged. Bars indicate the standard error of the mean. Different letters represent a significant difference at *P* < 0.05 (one-way ANOVA with a Fisher’s post hoc test).

**Figure 4 plants-08-00296-f004:**
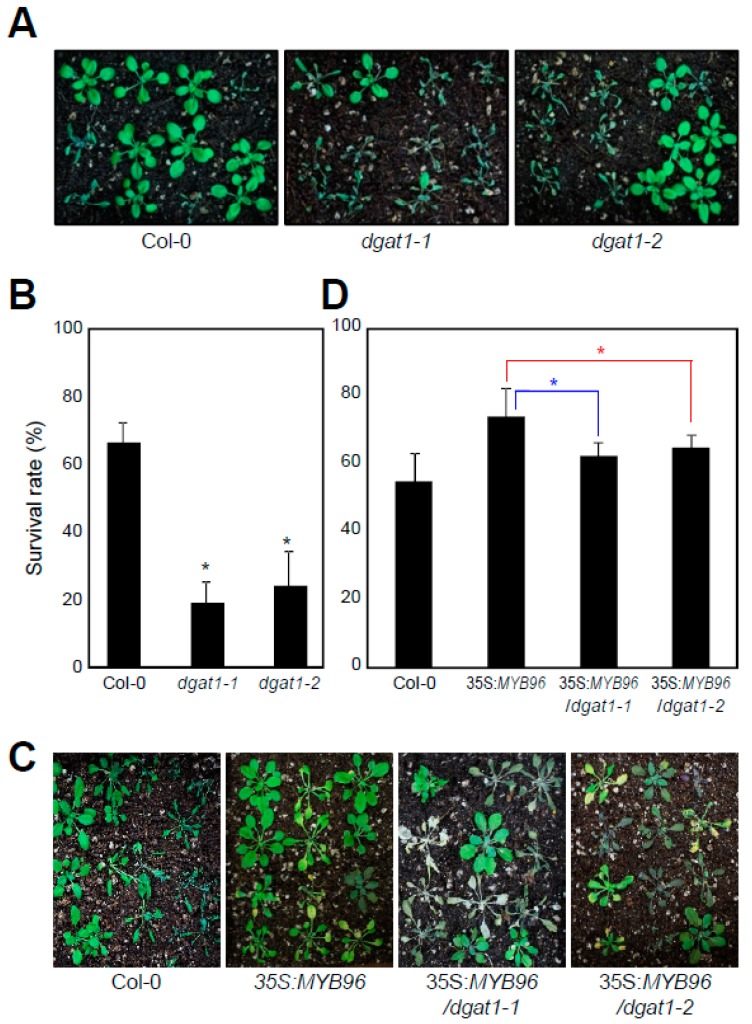
Drought-sensitive phenotypes of TAG-deficient mutants. Two-week-old plants were subjected to drought conditions by withholding water for an additional two weeks. At least five containers of multiple genotypes (30 plants/container) were evaluated in three independent experiments. In (**B**) and (**D**), plant survival rate was determined 3 d after rewatering. Three independent experiments with three biological triplicates were averaged and statistical significance of the measurements was determined using a Student’s *t*-test (* *P* < 0.05). Bars indicate the standard error of the mean. (**A**) Drought susceptibility of *dgat1* mutants. Photographs were taken 10 d after rewatering. (**B**) Survival rate of *dgat1* mutants. (**C**,**D**) Drought tolerance of 35S:*MYB96*/*dgat1* plants.

**Figure 5 plants-08-00296-f005:**
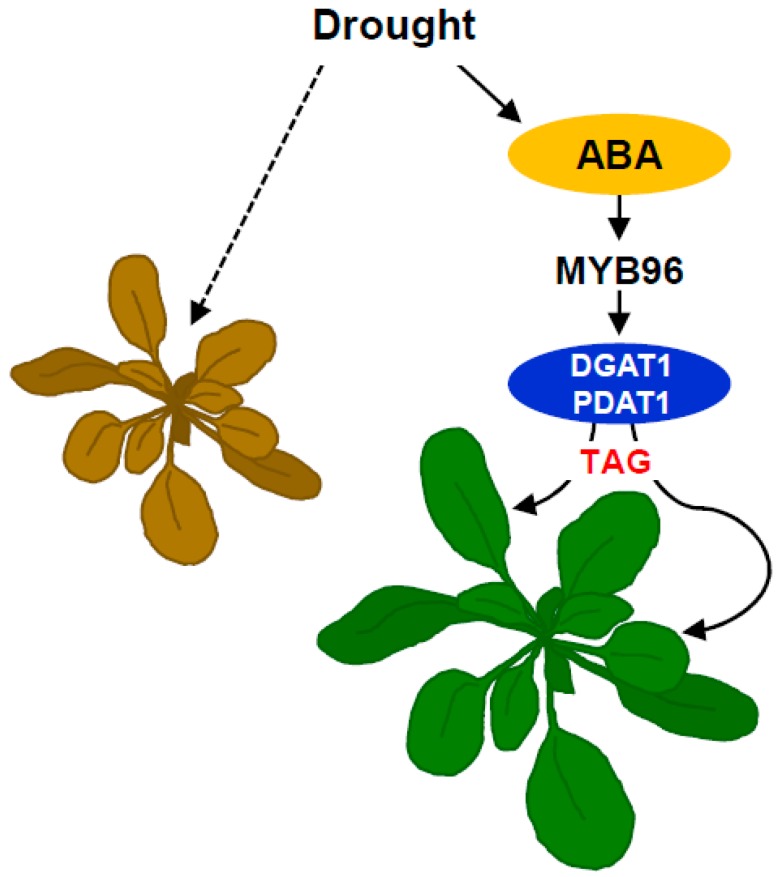
Working diagram. The ABA-inducible MYB96 transcription factor transcriptionally activates *DGAT1* and *PDAT1*, which catalyze TAG biosynthesis at the rate-limiting step. TAG accumulation in vegetative tissues confers drought tolerance and long-term adaptation to stressful conditions. Solid lines indicate the findings shown in this study. The dashed line indicates a result expected if MYB96-mediated signaling is lost.

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
