# Peer review of "The Arabidopsis MYB96 Transcription Factor Mediates ABA-Dependent Triacylglycerol Accumulation in Vegetative Tissues under Drought Stress Conditions"

_plants, 2019, doi:10.3390/plants8090296_

Round 1
Reviewer 1 Report
The manuscript titled “The Arabidopsis MYB96 transcription factor mediates ABA-dependent triacylglycerol accumulation in vegetative tissues under drought stress conditions” provides new insight into the molecular mechanisms how MYB96 plays important roles in response to adverse environments through biosynthesis of TAG. Authors showed that the amount of TAG is increased in the MYB96-overexpresssing plants, and decreased in the myb96-1 mutants treated with ABA. Accordingly, expression levels of DGAT1 and PDAT1, key enzymes for TAG biosynthesis, are upregulated in the MYB96-overexpressing plants, and suppressed by mutation of MYB96. Additionally, authors showed that dgat1 mutants were sensitive to drought stress. These results clearly indicate that regulation of TAG biosynthesis by MBY96 has critical roles to adapt to drought stress conditions, but this manuscript requires several revisions.
Major point
1. The important point which is missing in this manuscript is the data to show the effect of MYB96 on the expression levels of DGAT1 and PDAT1 and accumulation of TAG under drought stress conditions. Author should show those values in the myb96 mutants not only treated with ABA, but also treated with drought stress.
Minor points
1. The wording “ABA induction of TAG biosynthesis” is confusing. “ABA-inducible TAG biosynthesis” or “TAG biosynthesis induced by ABA” should be proper.
2. In the Materials and methods, the accession number of dgat1-1 should be shown.
3. Photographs of plants after drought stress associated with Figure 4C should be shown.
Author Response
Responses to Reviewer 1
The important point which is missing in this manuscript is the data to show the effect of MYB96 on the expression levels of DGAT1 and PDAT1 and accumulation of TAG under drought stress conditions. Author should show those values in the myb96 mutants not only treated with ABA, but also treated with drought stress.à Transcript accumulation of DGAT1 and PDAT1 was investigated, and the data were included in the revision (see Figures 2B-2D). However, we could not measure TAG accumulation in seedlings treated with drought stress, because substantial cell death occurred, making significant artifacts in the result. Please understand our situation.
Minor points
The wording “ABA induction of TAG biosynthesis” is confusing. “ABA-inducible TAG biosynthesis” or “TAG biosynthesis induced by ABA” should be proper.à Thank you for the suggestion. We revised the description as suggested.
In the Materials and methods, the accession number of dgat1-1 should be shown.
à The accession number was provided as suggested.
Photographs of plants after drought stress associated with Figure 4C should be shown.
à We included the representative photographs as suggested.
Reviewer 2 Report
Hello;
This manuscript by Lee et al. on The Arabidopsis MYB96 transcription factor mediates ABA-dependent triacylglycerol accumulation in vegetative tissues under drought stress conditions presented by authors for plants, is well written and covered methods, results, and conclusion appropriately. I suggest author to minor copy-edit the manuscript to get rid of typos, redundancy and clarity of a few sentences in the manuscript. And also in the Figure 2 labels are not explained (X axis and Y axis labels). Please explain them in legend.
Thank you.
Author Response
Responses to Reviewer 2
This manuscript by Lee et al. on The Arabidopsis MYB96 transcription factor mediates ABA-dependent triacylglycerol accumulation in vegetative tissues under drought stress conditions presented by authors for plants, is well written and covered methods, results, and conclusion appropriately. I suggest author to minor copy-edit the manuscript to get rid of typos, redundancy and clarity of a few sentences in the manuscript. And also in the Figure 2 labels are not explained (X axis and Y axis labels). Please explain them in legend.
à We reviewed the manuscript and revised mistakes and unclear descriptions. In addition, we also included the explanations about the figures in the Figure Legends.
Round 2
Reviewer 1 Report
The manuscript titled “The Arabidopsis MYB96 transcription factor mediates ABA-dependent triacylglycerol accumulation in vegetative tissues under drought stress conditions” provides new insight into the molecular mechanisms how MYB96 plays important roles in response to adverse environments through biosynthesis of TAG. Authors showed that the amount of TAG is increased in the MYB96-overexpresssing plants, and decreased in the myb96-1 mutants treated with ABA. Accordingly, expression levels of DGAT1 and PDAT1, key enzymes for TAG biosynthesis, are upregulated in the MYB96-overexpressing plants, and suppressed by mutation of MYB96 with ABA treatment and under osmotic, salt and dehydration stress. Additionally, authors showed that dgat1 mutants were sensitive to drought stress. These results clearly indicate that regulation of TAG biosynthesis by MBY96 has critical roles to adapt to drought stress conditions, and further analysis of roles of TAG under drought stress conditions is expected to be performed in future works.
Author Response
The manuscript titled “The Arabidopsis MYB96 transcription factor mediates ABA-dependent triacylglycerol accumulation in vegetative tissues under drought stress conditions” provides new insight into the molecular mechanisms how MYB96 plays important roles in response to adverse environments through biosynthesis of TAG. Authors showed that the amount of TAG is increased in the MYB96-overexpresssing plants, and decreased in the myb96-1 mutants treated with ABA. Accordingly, expression levels of DGAT1 and PDAT1, key enzymes for TAG biosynthesis, are upregulated in the MYB96-overexpressing plants, and suppressed by mutation of MYB96 with ABA treatment and under osmotic, salt and dehydration stress. Additionally, authors showed that dgat1 mutants were sensitive to drought stress. These results clearly indicate that regulation of TAG biosynthesis by MBY96 has critical roles to adapt to drought stress conditions, and further analysis of roles of TAG under drought stress conditions is expected to be performed in future works.à Thank you very much for your support. We would like to continue our work to further unravel the linkage between lipid metabolism and stress responses.